# Expression of Progesterone Receptor A as an Independent Negative Prognosticator for Cervical Cancer

**DOI:** 10.3390/ijms24032815

**Published:** 2023-02-01

**Authors:** Fabian Garrido, Carl Mathis Wild, Udo Jeschke, Christian Dannecker, Doris Mayr, Vincent Cavailles, Sven Mahner, Bernd Kost, Helene H. Heidegger, Aurelia Vattai

**Affiliations:** 1Department of Obstetrics and Gynecology, University Hospital Augsburg, 86156 Augsburg, Germany; 2Department of Data Management and Clinical Decision Support, Faculty of Medicine, University of Augsburg, 86159 Augsburg, Germany; 3Department of Obstetrics and Gynecology, University Hospital, LMU Munich, 81377 Munich, Germany; 4Department of Pathology, LMU Munich, 80337 Munich, Germany; 5Institut de Recherche en Cancérologie de Montpellier (IRCM), INSERM U1194, Université Montpellier, F-34298 Montpellier, France

**Keywords:** progesterone receptor A, cervical cancer, RIP140, HPV E6, p16, FIGO, negative prognosticator

## Abstract

The role of progesterone receptor A (PRA) for the survival outcome of cervical cancer patients is ambiguous. In mouse models, it has been shown that PRA plays a rather protective role in cancer development. The aim of this study was to assess its expression by immunohistochemistry in 250 cervical cancer tissue samples and to correlate the results with clinicopathological parameters including patient survival. PRA expression was positively correlated with the International Federation of Gynecology and Obstetrics (FIGO) classification scores. PRA was significantly overexpressed in adenocarcinomas compared to squamous epithelial carcinoma subtypes. Correlation analyses revealed a trend association with the HPV virus protein E6, a negative correlation with p16 and a positive correlation with EP3. PRA expression was also associated with the expression of RIP140, a transcriptional coregulator that we previously identified as a negative prognostic factor for survival in cervical cancer patients. Univariate survival analyses revealed PRA as a negative prognosticator for survival in patients with cervical adenocarcinoma. Multivariate analyses showed that simultaneous expression of RIP140 and PRA was associated with the worst survival, whereas with negative RIP140, PRA expression alone was associated with the best survival. We can therefore assume that the effect of nuclear PRA on overall survival is dependent upon nuclear RIP140 expression.

## 1. Introduction

Cervical carcinoma is the second most common female tumor entity worldwide, ranking third in terms of tumor-related cause of death [1,2]. The two main malignant epithelial cervical cancer types are squamous cell carcinoma and adenocarcinoma (about 70% and 10–25% of all cervical carcinomas, respectively) [3]. A persistent infection with high-risk human papillomavirus (HR-HPV) is the major leading cause of cervical cancer [4,5]. When HPV replicates, the viral E6 oncoprotein is expressed and disturbs the cell cycle [6,7,8]. The E6 oncoprotein and the E6-associated protein (E6-AP) form a complex which binds to the tumor suppressor protein p53 (an inducer of cell cycle arrest or apoptosis [9]) and causes its proteolytic degradation [9,10,11]. The cyclin-dependent kinase inhibitor p16ink4a (p16) is a tumor suppressor in most cells [12], but in cervical cancer with oncoprotein E7 expressed by HPV and the degradation of the retinoblastoma protein (Rb), p16 exhibits oncogenic activity [13].

Like other gynecologic malignancies, cervical cancer may also be influenced by female steroid hormones, predominantly estrogen and progesterone [14]. Nevertheless, the role of progesterone, in particular, in cervical cancer appears to be poorly understood to date. Recent studies in mice show a possible positive influence of progesterone receptor (PR) activating drugs on cancer cell growth [15]. Hong and colleagues were able to show a beneficial prognosis from increased progesterone receptor B (PRB) expression in an evaluation of tissue from a total of 95 patients with cervical cancer. Interestingly, a significantly higher expression of PRB was found in the cervical stroma as compared to the tumor itself [16].

Receptor-interacting protein 140 (RIP140) is a transcriptional coregulator of nuclear receptors, which controls hormone-induced gene transcription in numerous different tissues [17]. It plays a role in metabolic processes and promotes the expression of proinflammatory cytokines [15,18]. However, RIP140 also appears to play an important role in tumors. Xiao-Hong Yu and colleagues have shown that a high expression of RIP140 is associated with poorer survival in patients with triple negative breast cancer (TNBC). In addition, downregulation of RIP140 was shown to the limit growth and proliferation of mammary carcinoma cells in a mouse model [19]. Increased expression of RIP140 also appears to be associated with poorer survival in cervical carcinoma [20].

The aim of this study was to analyze the expression of progesterone receptor A (PRA) by immunohistochemical evaluation using a semiquantitative scoring system in a panel of 250 patients diagnosed with cervical cancer. Furthermore, we aimed to correlate PRA expression with the TNM classification; grading; histochemical subtype; FIGO (Fédération Internationale de Gynécologie et d’Obstétrique) staging and with various parameters that we previously quantified, including steroid hormone receptors; HPV proteins such as E6 and E7; p16; p53; and RIP140. In addition, the prognostic value of PRA was analyzed by univariate and multivariate analyses (Cox-regression).

## 2. Results

### 2.1. Expression of PRA in Patients with Cervical Cancer

Within the study, a total of 250 patients with cervical cancer were analyzed, with 80 percent of patients having a squamous cell carcinoma and 20 percent an adenocarcinoma. The median age of these patients was 47 years, ranging from 20 to 83 years. A summary of the clinical data is shown in Table 1.

In 127 cases (50.8%), the nuclear immunoreactivity score (IRS) of PRA was 0 or 1, which was considered as negative. By contrast, a total of 123 patients (49.2%) showed a positive expression of PRA with an IRS of two or more. As shown in Figure 1, when comparing the histological subtypes, there was a significant (*p* = 0.044) stronger expression of nuclear PRA in patients with adenocarcinoma (mean IRS = 1) compared to the subgroup with squamous cell cancer (mean IRS = 2).

Cytoplasmic PRA staining was considered negative in 237 cases (94%), with 222 cases (88%) with an IRS of zero and 15 patients (6%) with an IRS of one. In 13 cases (5.2%), the IRS of cytoplasmic PRA was at least two. The level of cytoplasmic PRA was not significantly different in the two types of cervical cancers.

### 2.2. Correlation Analysis of PRA and RIP140

Nuclear PRA expression in cervical cancer patients (*n* = 123) was compared with patients without nuclear PRA expression (*n* = 127), showing that high nuclear PRA expression was associated with a high expression of cytoplasmic RIP140 (*p* = 0.019), as shown in Table 2A,B. We also saw a connection with simultaneous nuclear RIP140 expression without reaching statistical significance (*p* = 0.066). Stained samples for simultaneous and single PRA and RIP140 expression are shown as examples in Figure 2. A corresponding tendency in the comparison of cytoplasmic PRA and RIP140 expression was not shown.

When analyzing PRA expression with other histological markers, there was a significant positive correlation between nuclear PRA expression and cytoplasmic expression of E6. We also noticed a significant inverse correlation between nuclear PRA and cytoplasmic p16.

Further, a subgroup analysis was performed for patients with adenocarcinoma. Here, only a positive correlation between nuclear PRA expression and EP3 and a significant inverse correlation between nuclear PRA and cytoplasmic p16 could be detected. This is shown in Table 2A,B.

### 2.3. Survival Analysis

Survival analysis was then performed after subdividing patients by histologic type. In the squamous cell carcinoma group (*n* = 199), a total of 41 patients (20.6%) showed high nuclear expression of PRA (IRS ≥ 4). However, with a *p*-value of 0.457, there was no statistically significant correlation with patient survival.

In contrast, in the adenocarcinoma group (*n* = 48), 11 patients (22.9%) showed strong nuclear PRA expression (IRS ≥ 4). Survival analysis showed a significantly worse overall survival for patients with high expression of PRA (*p* = 0.012), which is shown in Figure 3.

Since RIP140, a modulator of PR expression and activity [21], is known to have a prognosis value in cervical cancers [20], we tested whether combining PRA and RIP140 expression could improve the correlation with survival in adenocarcinoma patients. In the group of patients with adenocarcinoma, 8 (17.0%) of the 47 patients showed simultaneous expression (IRS ≥ 4) of nuclear RIP140 and PRA. In total, 25 patients (53.2%) showed expression of RIP140 only, while 12 patients (25.5%) expressed neither PRA nor RIP140. Two (4.3%) patients showed increased expression of PRA only (IRS ≥ 4). As shown in Figure 4, survival analysis showed the best outcome for the latter group, while patients expressing both PRA and RIP140 simultaneously showed the worst overall survival (*p* = 0.009).

As shown in Figure 5, the comparison of the patients simultaneously expressing PRA and RIP140 (*n* = 8) to all other patients (*n* = 39) within the adenocarcinoma group (*n* = 47) showed a highly significant trend (*p* < 0.001).

Since the results shown in Figure 4 and Figure 5 suggested that RIP140 might influence the prognosis value of PRA, we then investigated if PRA expression was differentially correlated with survival in the subgroups of adenocarcinoma patients having either low or high expression of RIP140 (Figure 6). In order to homogenize the number of patients in each arm, the cutoff for PRA was set at IRS ≥ 2.

Amongst the 28 tumors expressing high levels of RIP140 (IRS ≥ 4), no significant correlation (*p* = 0.738) of PRA expression with survival was observed. As shown in Figure 6A, the 16 patients which expressed PRA (IRS ≥ 2) did not exhibit differences in term of survival when compared to the 12 patients which expressed PRA at low levels (IRS < 2).

By contrast, Figure 6B shows that amongst the 14 tumors expressing low levels of RIP140 (IRS < 4), a significantly better survival (*p* = 0.041) for the 7 patients with tumors expressing PRA as compared to the 7 patients expressing PRA at low levels (IRS < 2). Altogether, these data clearly support the hypothesis that RIP140 expression influences the prognosis value of PRA in patients with cervical adenocarcinoma. In term of patient survival, a high expression of PRA might be deleterious if RIP140 is co-expressed and beneficial when RIP140 levels are the lowest.

### 2.4. Cox Regression Analysis

Cox regression analysis was performed after subdivision into histologic subtypes. No significant independent risk factors emerged in the squamous cell carcinoma cohort.

In the adenocarcinoma group, concurrent nuclear expression of PRA (IRS ≥ 4) and RIP140 emerged as an independent risk factor for overall survival (*p* = 0.037). The same could not be shown for TNM status, age or FIGO classification. These results are shown in Table 3.

## 3. Discussion

Within this study, we have shown that the combined nuclear expression of the progesterone receptor A (PRA) and RIP140 (receptor interacting protein of 140 kDa), also known as NRIP1 (nuclear receptor interacting protein 1), is an independent negative prognosticator for patients with cervical adenocarcinomas. In addition, high expression of PRA alone is a negative prognosticator in univariate analyses. In the course of further subdivision of tumor entities according to histological types, nuclear expression of PRA (IRS ≥ 4) was initially shown to be a negative prognostic marker for patients with adenocarcinoma, which could not be presented in this way for squamous cell carcinoma. However, in combination with nuclear RIP140 expression, the cohort of adenocarcinomas again showed a different behavior of PRA on overall survival. Thus, simultaneous expression of RIP140 and PRA was associated with the worst survival, whereas with negative RIP140, PRA expression alone was associated with the best survival (outlined in Figure 4 and Figure 5). With a cutoff of IRS ≥ 4, the cohort of only PRA expressing adenocarcinoma was very small at *n* = 2. To increase the number of cases, further analyses with a cutoff of IRS ≥ 2 were performed, again confirming that PRA expression with RIP140 negativity is associated with a better prognosis. This suggests that nuclear RIP140 expression determines the effect of PRA on overall survival.

In their work, Vattai and colleagues demonstrated that nuclear RIP140 expression is associated with significantly worse survival in patients with cervical cancer. However, this was only seen in squamous histology [20]. In a recent study by Madak-Erdogan and colleagues, it was shown that nuclear RIP140 bound to the estrogen complex can block the estrogen receptor and thus prevent estrogen-mediated gene expression [22]. If RIP140 behaves similarly with nuclear expression with respect to progesterone receptor A, this may explain the contrasting effect of PRA dependent RIP140 expression on overall survival of patients with adenocarcinoma of the uterine cervix.

The progesterone receptor is known as a positive marker for overall survival of different gynecologic cancer types. A former study of our group showed that the loss of PRA and PRB resulted in poorer survival in endometrial cancer patients [23]. In addition, similar results could be obtained with ovarian cancer. We showed that progesterone receptor B is a positive prognostic marker for cervical cancer patient survival [24]. In addition, concurrent expression of cytoplasmic NRF2 and PRA/PRB were associated with a significantly longer OS [25]. In ovarian cancer, progesterone plays an anti-proliferative effect via its receptor and has hereby been reported to be associated with improved OS and PFS patients [26,27]. These findings are supported by studies showing that PR mediates apoptotic cell death [28]. Furthermore, upregulation of forkhead box transcription factor (FOXO1) through progestin activated PRA and PRB causes cell cycle arrest by the increase of mediators of cell senescence [29].

In breast cancer, the role of PR is ambivalent. A former study showed that PRA has been associated with a poor clinical outcome with more rapid disease recurrence after tamoxifen treatment [30]. In addition, transgenic mice with an excess of PRA are characterized by disproportionate lateral ductal branching [31]. A recent study showed that methylation of PRA but not PRB is predictive for tamoxifen response; therefore, the authors assume that an association between PR promoter methylation and worse outcome in breast cancer patients exists [32]. In addition, it could be shown that PRA inhibited gene expression and ER chromatin binding significantly more than PRB. Differential gene expression was observed in PRA and PRB-rich breast cancer tissues, and PR-A-rich gene signatures had poorer survival outcomes [33]. Finally, the expression of the immunosuppressive protein glycodelin A in breast cancer correlates with PRA [34]. A similar ambivalence was found in our study, demonstrating the importance of analyzing further co-factors.

Based on the correlation analyses of this study, we identified a variety of proteins that are positively correlated with PRA based on our previous publications: HPV-E6, EP-3 and RIP140. A negative correlation with PRA could be detected with p16. The G-protein coupled prostaglandin receptor EP3 was already identified as a negative prognosticator for cervical cancer [35]. The combined survival analyses of PRA and EP3 did not result in new findings. Similar negative results were obtained with combined E6 and PRA analyses [7,14]. However, the analysis of simultaneous expression of RIP140 as well as PRA yielded new findings. Interestingly, PRA showed a positive correlation with the G-protein coupled estrogen receptor but not with the glucocorticoid receptor (GR) [36]. There was a positive correlation between GR and RIP140 based on our results [20]. In former studies, it could be shown that RIP140 can act as a corepressor for GR and represses GR activation of a glucocorticoid response element (GRE)-regulated reporter gene [37].

Up to date, only a few investigations exist about the interaction of RIP140 and PR. Investigations with a RIP140-KO model showed that the knockout of RIP140 inhibits the expression of PR by almost 100% [21]. Interestingly, this effect was not seen for ERα [21]. Therefore, we must consider, that RIP140 directly regulates PR expression. Within this study, we found that 41.2% of all cases are negative for nuclear PRA expression. A total of 37.6% of the patients showed a low expression (IRS 1 to 3) and 20.3% of the cases showed an elevated expression (IRS ≥ 4) of PRA. A recent study summarized that in 60–80% cases, the progesterone receptor in the tissue of cervical carcinoma is not detectable [38]. These numbers are almost in line with our immunohistochemical study of PRA in cervical cancer. A limitation of our study might be that we only investigated the expression of PRA but not PRB in the samples. This seems to be an important starting point for further investigations.

The current evaluation was carried out with cervical cancer samples from 1993 to 2002. In the meantime, therapy options for cervical cancer patients have been modified which can further have an influence on the follow-up period. Even though this study is a monocentric design, the collective is very large with tissue samples from a total of 250 patients with cervical cancer. The correlations we have shown can serve as a basis for subsequent, possibly multicenter studies, which first and foremost evaluate the clinical utility of our findings.

## 4. Materials and Methods

### 4.1. Patient Collective

For this study, we included formalin fixed paraffin embedded cervical cancer samples of 250 patients (without distant metastasis), that underwent surgery in the years from 1993 to 2002 at the Department of Gynecology and Obstetrics, Ludwig-Maximilians-University, Munich, Germany. This happened without any preselection. Only histological subtype squamous cell carcinoma and adenocarcinoma participated in the cohort. The data, including clinical and follow-up data such as patient age, OS, lymph node status, tumor size, presence of metastases, histopathological grading, tumor subtype and FIGO stage, were retrieved from the Munich Cancer Registry.

### 4.2. Immunohistochemistry

The expression of progesterone receptor A was immunohistochemically quantified from the embedded samples. At first, dewaxing was performed by Roticlear (Carl Roth GmbH, Grafenrath, Germany), and the sections were then rehydrated in 100% ethanol. Endogenous peroxidase was blocked by using 3% hydrogen peroxide in methanol (=3 mL 30% H_2_O_2_ + 97 mL methanol). Sections were swirled another time in ethanol (100%, 70% and 50%) as well as in water before demasking was performed by a pressure cooker with a Na citrate buffer with a pH of 6.0. The buffer solution was mixed as follows: Solution A: 21.01 g 0.1 M citric acid (Merck#244) + 1 L distilled water. Solution B: 29.41 g 0.1 M Na citrate (Merck#6448, Darmstadt, Germany) + 1 L distilled water. Usage lsg: 18 mL lsg.A + 82 mL lsg.B + 900 mL distilled aqua. Again, sections were washed in distilled water and phosphate-buffered saline (PBS). Blocking solution (reagent 1) was added for 5 min to avoid nonspecific staining. The tissue was then incubated with the primary antibody for progesterone receptor A (polyclonal rabbit IgG, Sigma-Aldrich, St. Louis, USA) for 16 h at 4° Celsius in a dilution of 1:250 with PBS. This was followed by two washing steps with PBS as well as an addition of post block reagent 2 for 20 min. The next step was adding HRP polymer (reagent 3) for 30 min and another washing step in PBS. Two minutes of substrate staining with DAB containing 1 mL substrate buffer and 1 drop of DAB chromogen. To stop the color reaction, washing in distilled water followed. Acidic hemalaun according to Mayer was added for counterstaining for 2 min. The tissue was blued for 5 min in tap water. Subsequently, alcohol was added in ascending series up to Roticlear. Finally, the samples were covered with “Roti-Mount”.

We then correlated our results with the former staining of these samples for E6, LCor, EP3, p16 and RIP140, which had recently been published [7,20,39].

### 4.3. Signal Evaluation (Immunoreactive Score)

The samples were analyzed by two observers using a Leitz Diaplan microscope (Leitz, Wetzlar, Germany). For the evaluation of the staining, the immunoreactive score (IRS) was used. The IRS scoring system reaches from 0 to 12. To obtain the IR score results, the staining intensity (score 0 = no staining, score 1 = weak staining, score 2 = moderate staining and score 3 = strong staining) and the percentage of positively stained cells (0: no staining, 1:  ≤ 10% of the cells, 2: 11–50% of the cells, 3: 51–80% of the cells and 4:  ≥ 81% of the cells) were multiplied. Cytoplasmic and nuclear signal quantification was performed with separate determination. An IR score greater than 1 was scored as positive.

### 4.4. Ethical Approval

The tissue samples used in this study were left over material after all diagnostics had been completed and were retrieved from the archive of the Department of Gynecology and Obstetrics, Ludwig-Maximilians-University, Munich, Germany. All patients gave their informed consent for additional research before undergoing surgery. The performed procedures were in accordance with the Declaration of Helsinki, 1975. All information and data of the patients were fully anonymized and afterwards encoded for further statistical analysis. This study was approved by the Ethics Committee of the Ludwig-Maximilians-University, Munich, Germany.

### 4.5. Statistical Analysis

For statistical analysis, the IBM Statistical Package for the Social Sciences (IBM SPSS Statistic v24.0 Inc., Chicago, IL, USA) was used. Survival times were compared using a Kaplan–Meier analysis. Correlation analysis was performed using Spearman’s Rho correlation. The Mantel–Cox log-rank test was used for the differences in overall survival. Not parametrical tests such as the Kruskal–Wallis test or Mann–Whitney U test were performed for comparisons of different groups.

A *p* value < 0.05 was considered to be significant. The *p* value and the number of patients analyzed in each group are given for each chart.

## 5. Conclusions

PRA is expressed in cervical cancer patients with an IRS ≥ 4 in only 20.3% of the patients, whereby adenocarcinomas show a higher expression than squamous cell carcinomas. For this histologic subtype, patients with PRA expression had a reduced overall survival compared to patients without PRA expression in the adenocarcinoma subgroup. However, on closer inspection, nuclear RIP140 expression appears to trigger the actual effect of PRA on overall survival, with its expression making PRA a negative prognostic marker. Analysis of PRA together with the transcriptional coregulator RIP140 may be a promising target for cervical cancer patients.

## Figures and Tables

**Figure 1 ijms-24-02815-f001:**
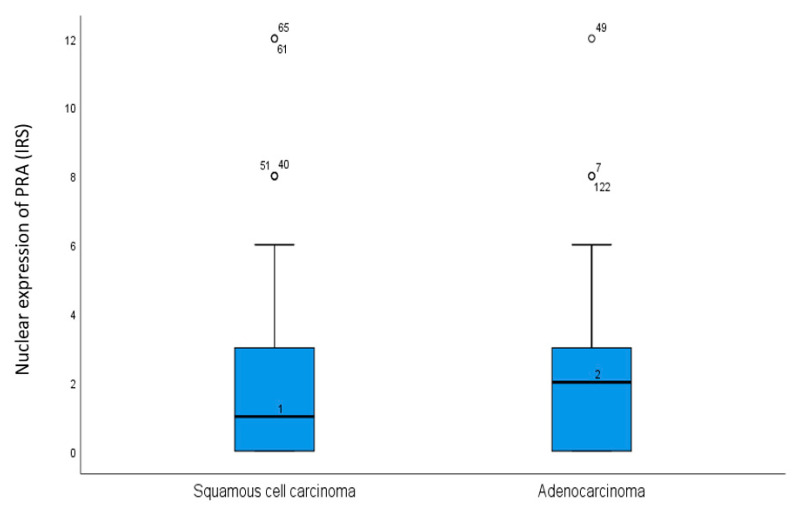
Boxplot comparing PRA expression in the two histological carcinoma types. There was a stronger nuclear expression of PRA in adenocarcinoma with a median of IRS 2 compared to squamous cell carcinoma with a median of IRS 1 (*p* = 0.044). The numbers represent outliers, and the circles represent outlier cases.

**Figure 2 ijms-24-02815-f002:**
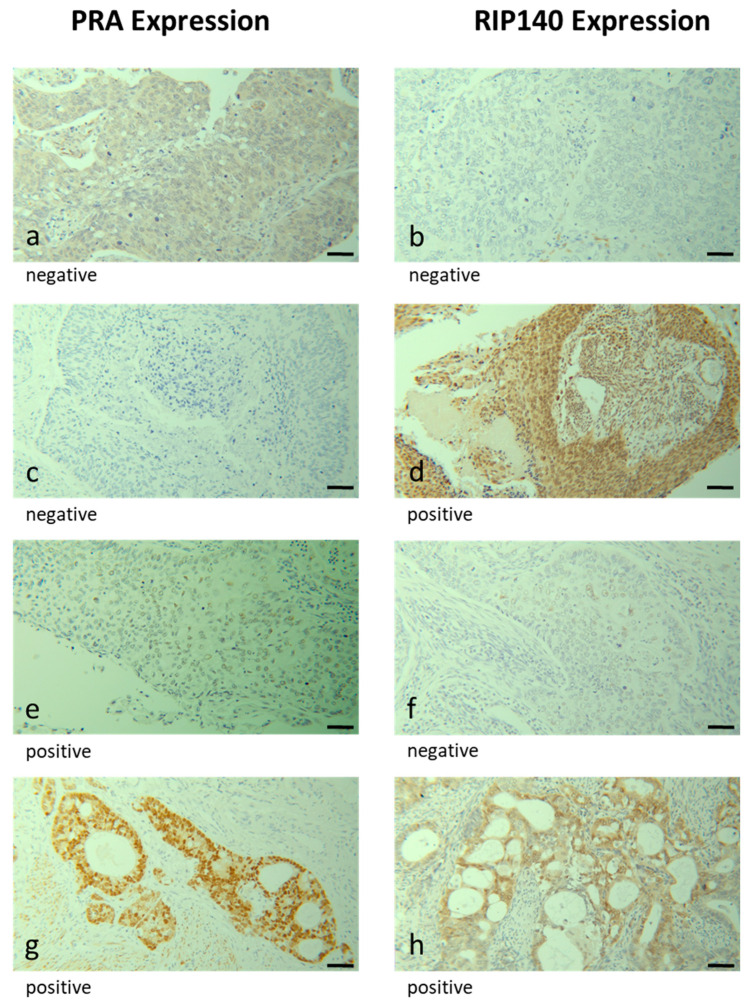
Immunohistochemical staining of PRA and RIP140. Immunohistochemical staining of PRA and RIP140 in human cervical cancer tissue is illustrated in (**a**–**h**): case 1 (**a**,**b**) negative expression of PRA and RIP140, case 2 (**c**,**d**) negative expression of PRA and positive expression of RIP140, case 3 (**e**,**f**) positive expression of PRA and negative expression of RIP140 and case 4 (**g**,**h**) positive expression of PRA and RIP140. The bar represents 100 µm.

**Figure 3 ijms-24-02815-f003:**
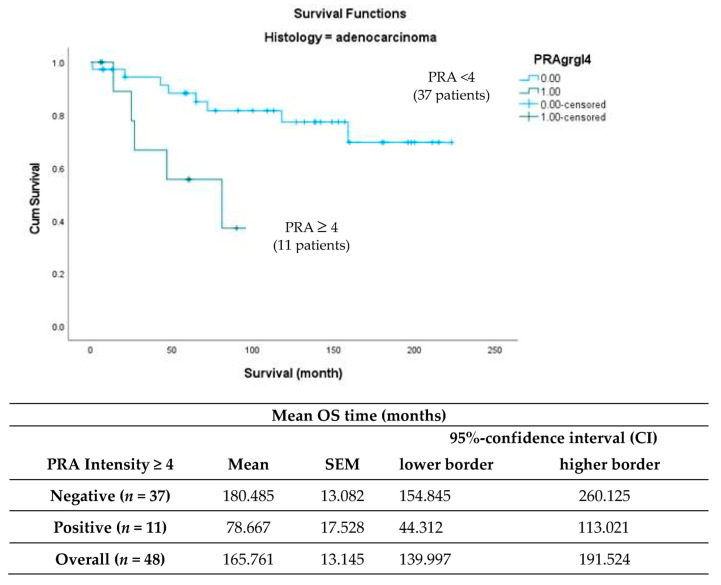
Kaplan–Meier survival analysis of high expression of progesterone receptor A in relation to overall survival (OS). Statistical significance is shown as *p*-value from log-rank test (*p* = 0.012). PRA = progesterone receptor A, IRS = immunoreactivity score and SEM = standard error of the mean.

**Figure 4 ijms-24-02815-f004:**
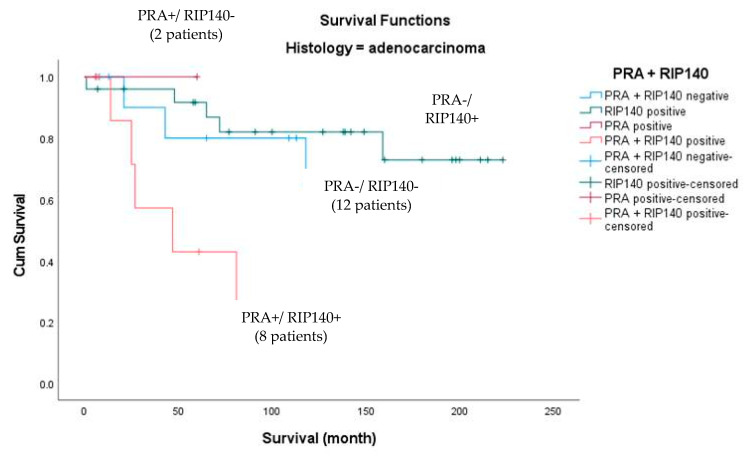
Kaplan–Meier survival analysis of nuclear RIP140 positive and progesterone receptor A high expression (IRS ≥ 4) in relation to OS for the subgroup of adenocarcinoma. Statistical significance is shown as *p*-value from log-rank test (*p* = 0.009). Mean +/− confidence interval of survival times for each group are not computed because all cases were censored.

**Figure 5 ijms-24-02815-f005:**
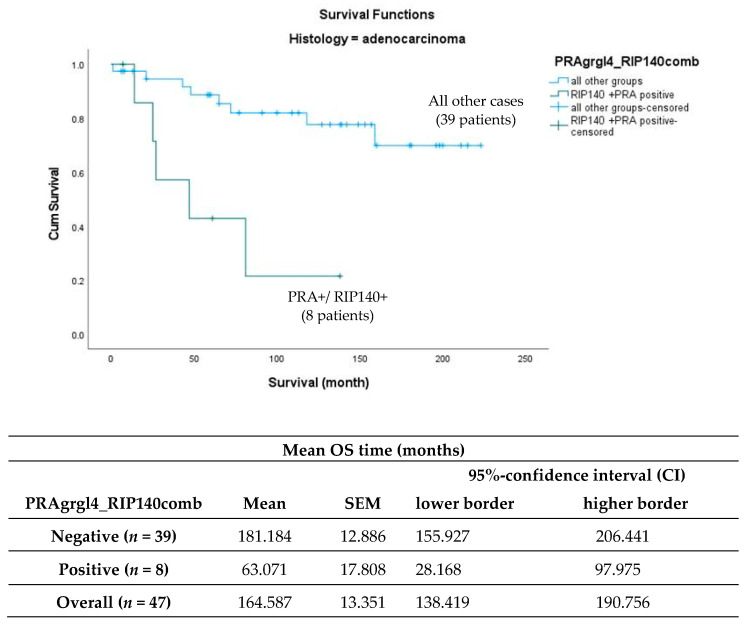
Kaplan–Meier survival analysis of patients with nuclear RIP140 and PRA high expression (IRS ≥ 4) compared to all other patients in relation to OS for the subgroup of adenocarcinoma. Statistical significance is shown as *p*-value from log-rank test (*p* < 0.001).

**Figure 6 ijms-24-02815-f006:**
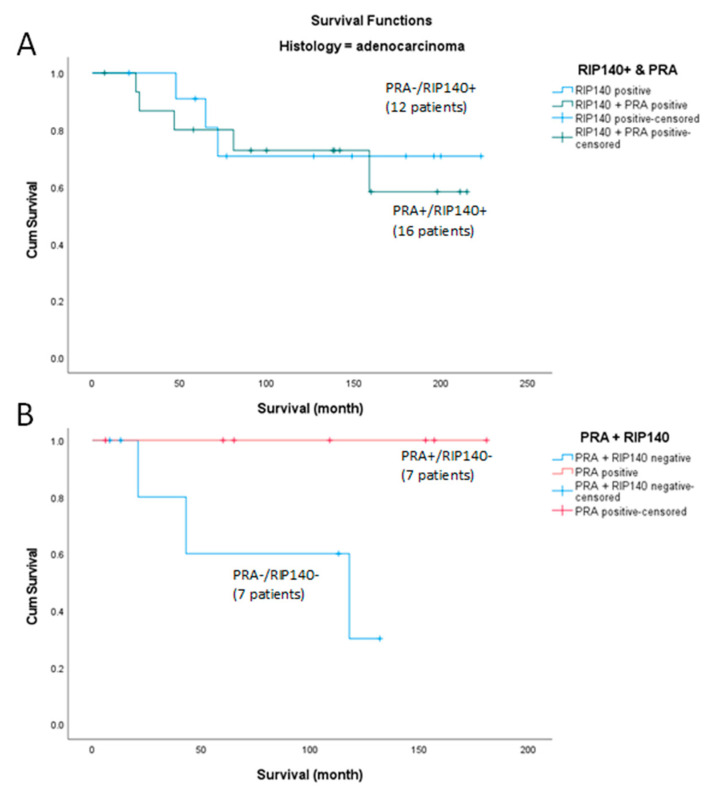
Kaplan–Meier survival analysis of adenocarcinoma patients according to nuclear PRA expression (cutoff IRS ≥ 2) in tumors with high levels of RIP140 (Panel (**A**), *p* = 0.738) or low levels of RIP140 (panel **B**). Statistical significance is shown as *p*-value from log-rank test for panel (**B**) (*p* = 0.041). Mean +/− confidence interval of survival times for each group are not computed because all cases were censored.

**Table 1 ijms-24-02815-t001:** Patient characteristics. IRS = immunoreactivity score, RIP140 = receptor-interacting protein 140 and PRA: progesterone receptor A.

Patient Characteristics	*n* (%)
Age (years)	Median 47.0 years 20–83
Histology	Squamous: 202/250 (80.8) Adenocarcinoma 48/250 (19.2)
Tumor grade	G1 21/250 (8.4) G2 143/250 (57.2) G3 78/250 (31.2)
pT	pT1 110/250 (44.0) pT2 128/250 (51.2) pT3/4 9/259 (3.6)
pN	pN0 151/250 (60.4) pN1 99 (39.6)
pM	pM0 250/250 (100) pM1 0/250 (0)
Nuclear RIP140	IRS = 0: 23/240 (9.6) IRS ≥ 1: 217 (90.4)
Cytoplasmic RIP140	IRS = 0: 21/240 (8.8) IRS ≥ 1: 219 (91.2)
Nuclear PRA	IRS ≤ 1: 127/250 (50.8) IRS > 1: 123/250 (49.2)
Cytoplasmic PRA	IRS ≤ 1: 237/250 (94.8) IRS > 1: 13/250 (5.2)

**Table 2 ijms-24-02815-t002:** (**A**) Significant correlation with nuclear PRA expression for all histologic types. (**B**) Correlation with nuclear PRA expression for adenocarcinoma.

**A**		
**Variables**	***p* Value**	**Correlation Coefficient**
RIP140 Cytoplasmic IRS	0.019	0.152
RIP140 Nuclear IRS	0.066	0.119
E6 Cytoplasmic IRS	0.024	0.144
P16 Cytoplasmic IRS	0.022	−0.149
EP3 IRS	0.000	0.236
**B**		
**Variables**	***p* Value**	**Correlation Coefficient**
RIP140 Cytoplasmic IRS	0.214	0.185
RIP140 Nuclear IRS	0.359	0.137
E6 Cytoplasmic IRS	0.509	0.098
P16 Cytoplasmic IRS	0.003	−0.425
EP3 IRS	0.038	0.303

**Table 3 ijms-24-02815-t003:** Multivariate Cox regression analysis regarding overall survival for adenocarcinoma.

Variable	Coefficient	HR (95%CI)	*p* Value
**pN**	0.987	2.683 (0.727–9.894)	0.138
**pM**	−0.112	0.894 (0.239–3.345)	0.868
**pT**	0.575	1.777 (0.952–3.316)	0.071
**FIGO**	−0.058	0.943 (0.826–1.077)	0.389
**Age**	−0.011	0.989 (0.935–1.045)	0.691
**PRA ≥ 4 + RIP140**	2.024	7.571 (1.135–50.518)	**0.037**

## Data Availability

The data presented in this study are available on request from the corresponding author. The data are not publicly available due to ethical issues.

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
