# Peer review of "Expression of Progesterone Receptor A as an Independent Negative Prognosticator for Cervical Cancer"

_ijms, 2023, doi:10.3390/ijms24032815_

Round 1

Reviewer 1 Report

The aim of this manuscript is to investigate the role of progesterone as an immunohistochemical marker useful in defining the prognosis in patients with cervical cancer. the authors state that PR was significantly overexpressed in adenocarcinomas compared to squamous epithelial carcinoma subtypes. Where's the news? This has already been amply demonstrated. In the opinion of this reviewer, it is conceptually wrong to mix two diseases (adenocarcinomas and squamous cell carcinomas) which in themselves have already known different biological and immunophenotypic characteristics. The role of progesterone should be studied in subpopulations of the same disease.

Author Response

Reviewer 1:

Open Review

The aim of this manuscript is to investigate the role of progesterone as an immunohistochemical marker useful in defining the prognosis in patients with cervical cancer. The authors state that PR was significantly overexpressed in adenocarcinomas compared to squamous epithelial carcinoma subtypes. Where's the news? This has already been amply demonstrated. In the opinion of this reviewer, it is conceptually wrong to mix two diseases (adenocarcinomas and squamous cell carcinomas) which in themselves have already known different biological and immunophenotypic characteristics. The role of progesterone should be studied in subpopulations of the same disease.

According to the point of criticism, the paper was reviewed again from the point of view of histological subdivision of squamous cell carcinoma and adenocarcinonoma of the uterine cervix. All analyses, especially the survival analyses, were performed again according to the appropriate subdivision. It was found that the influence of progesterone receptor A and RIP140 was significant only in the case of adenocarcinomas, whereas similar correlations could not be shown for squamous cell carcinomas.

Reviewer 2 Report

Although the authors  have done good job by studying the role of the progesterone receptor (PR) in cervical cancer patients at their settings, which is ambiguous but not critical.

However, addressing the following comments will make the manuscript comprehensive.

1. Using high resolutions pictures/graphs in figures and making figures better will be good.

2. Adding the patient number(n) in survival curves/graphs and adding the p values or 95% CI will be good.

3. Adding more relevant introduction and explaining the rationale to do the study will make manuscript all-inclusive and adding information on the "role on targeting the progesterone receptor (PR) in cervical cancer patients."

4. Add more details on sample collection which are 10-15 years old. Explain limitations and bias in using old saved samples.

Author Response

Reviewer 2:

Although the authors  have done good job by studying the role of the progesterone receptor (PR) in cervical cancer patients at their settings, which is ambiguous but not critical.

However, addressing the following comments will make the manuscript comprehensive.

  1. Using high resolutions pictures/graphs in figures and making figures better will be good.

The graphics have been improved in terms of presentation as well as captions. The same was done with regard to the inserted images.

  1. Adding the patient number(n) in survival curves/graphs and adding the p values or 95% CI will be good.

Patient numbers and p-values were added to all graphs and curves, respectively.

  1. Adding more relevant introduction and explaining the rationale to do the study will make manuscript all-inclusive and adding information on the "role on targeting the progesterone receptor (PR) in cervical cancer patients."

The introduction was completed regarding the role of PR as well as RIP140 and their influence on carcinomas.

  1. Add more details on sample collection which are 10-15 years old. Explain limitations and bias in using old saved samples.

We are fully aware of this fact but the advantage of this collection is the possibility to add a new evaluation to the already existing staining results. Only this gave us the possibility to analyze the prognostic value of PRA in comparison to the RIP140 expression and it led to the result that RIP140 determines the prognostic value of PRA at least in adenocarcinomas.  

Reviewer 3 Report

Review of “ Expression of progesterone receptor as a negative independent prognosticator for cervical cancer” by Garrido and colleagues .

In this paper the authors explored the role of the progesterone receptor (PR) and RIP140 for the survival outcome of cervical cancer using immunohistochemistry. Overall, I found the manuscript interesting and of relevance to researchers in the field. However, I do have some questions and comments which I believe should be addressed. I hope my comments will be of helpful for the authors to improve their work.

Detailed comments:

1)     Figure 1. The data presentation is not clear. What are the numbers #1 and #2 inside the boxes mean ? From the text it looked like they label IRS 1 vs IRS 2, but IRS levels in the graph are different. I suggest deleting it, and to avoid labeling groups by numbers but rather by name of the group.

2)     Also Figure 1, few data points (the extremely high) are labeled by numbers, what are these number ? sample IDs ? please explain.

3)     Figure 2: Scale bars are missing in all images. Also, letters that labeled the images partly overlap the images .

4)     Table 1 .  I suggest , to show the data and its distribution. Scatter plots would be much more helpful than a summary table to visualize and inspect the data.

5)     Table 1. It is not clear how was the correlation analysis done. Was it linear or not ? Is it a Pearson or Spearman correlation ? This should be also detailed in the methods section, in the stat sub-section.

6)     Many abbreviations were not explained, or explained only at the end of the paper in the methos section, they and might be unclear to a wider readership . Examples: IRS, PR receptors are abbreviated as: PR; PR-A; PRA; PR-B. This is probably for the different isoform of the PgR gene but should be explained and written in a consistent way in along the text. OS- overall survival etc…

7)     Tables 3-4 and Kaplan-Meier plots. It is not clear why the authors chose their cutoff at >7 and >4. Why to change the cutoff between analyses. The sample sizes for this cutoff is very small with n=6-8. One would believe a consistent cutoff would be better, and maybe a lower cutoff would increase the sample size. Is there a clinical reasoning for this ? If so, please detail it and please justify the cutoff and small sample sizes of the subgroups.

8)     Table 4. Line labeled as “ only one”. I believe having two lines : only PR and only RIP140 , would describe the results better. Is there a reason to pull them together ?

9)     Discussion : “In addition, similar results could be obtained in ovarian cancer.”  Citation needed.

10)  Methods: “We then correlated our results with the former staining of these samples for E6, LCor, RIP140, nuclear p53, H3K9ac and H3K4me3, which had recently been published” Correlation analysis with PR and RIP140 for some of those measures is not shown in the paper (H3K9ac, H3K4me3, P53,,,).

11)  Conclusion: “The analyses of PR together with the transcription co-regulator RIP140 could act as a promising target not only for cervical cancer patients”   When saying “not only” please explain and give an example.

12)  What is the possible biological explanations for PR levels being positive vs negative markers in different gynecological cancers ?

13)  The immunohistochemistry was done with an antibody against PR-A but not against PR-B. It is, at least theoretically,  possible that PR-B levels compensate for the PR-A levels which can affect the overall conclusions of this study. This should be addressed and mentioned as a possible limitation of the study.

Author Response

Reviewer 3:

Review of “ Expression of progesterone receptor as a negative independent prognosticator for cervical cancer” by Garrido and colleagues .

In this paper the authors explored the role of the progesterone receptor (PR) and RIP140 for the survival outcome of cervical cancer using immunohistochemistry. Overall, I found the manuscript interesting and of relevance to researchers in the field. However, I do have some questions and comments which I believe should be addressed. I hope my comments will be of helpful for the authors to improve their work.

Detailed comments:

  • Figure 1. The data presentation is not clear. What are the numbers #1 and #2 inside the boxes mean ? From the text it looked like they label IRS 1 vs IRS 2, but IRS levels in the graph are different. I suggest deleting it, and to avoid labeling groups by numbers but rather by name of the group.

The table was changed, the IRS was adjusted to the cut off of the text below.

  • Also Figure 1, few data points (the extremely high) are labeled by numbers, what are these number ? sample IDs ? please explain.

The numbers represent outliers, and the circles represent outlier cases. A note to this effect has been added to the graphic.

  • Figure 2: Scale bars are missing in all images. Also, letters that labeled the images partly overlap the images.

Scale bars have been added, and the labels have been optimized accordingly.

  • Table 1. I suggest , to show the data and its distribution. Scatter plots would be much more helpful than a summary table to visualize and inspect the data.

Because we published clinical data in the other cervical cancer publications of our group [1-20], we would like to keep the data distribution in form of a table, but we fully agree that the presentation in scatter plots would be a good alternative.

  • Table 2. It is not clear how was the correlation analysis done. Was it linear or not? Is it a Pearson or Spearman correlation ? This should be also detailed in the methods section, in the stat sub-section.

Correlation analysis was performed using Spearman's Rho correlation. This was added under Methods.

  • Many abbreviations were not explained, or explained only at the end of the paper in the methos section, they and might be unclear to a wider readership . Examples: IRS, PR receptors are abbreviated as: PR; PR-A; PRA; PR-B. This is probably for the different isoform of the PgR gene but should be explained and written in a consistent way in along the text. OS- overall survival etc…

In this regard, the paper was reviewed again, abbreviations were explained accordingly, moreover, they were used consistently.

  • Tables 3-4 and Kaplan-Meier plots. It is not clear why the authors chose their cutoff at >7 and >4. Why to change the cutoff between analyses. The sample sizes for this cutoff is very small with n=6-8. One would believe a consistent cutoff would be better, and maybe a lower cutoff would increase the sample size. Is there a clinical reasoning for this? If so, please detail it and please justify the cutoff and small sample sizes of the subgroups.

Data were reprocessed by histologic type, and therefore new survival curves were inserted. A continuous cut off was chosen (IRS >=4). In case of very small subgroups, additional analyses with other cut off values were performed in order to achieve a greater significance due to more patients. This has now been explained accordingly in the text.

  • Table 4. Line labeled as “only one”. I believe having two lines: only PR and only RIP140 , would describe the results better. Is there a reason to pull them together ?

This graphic was also removed and replaced by others after renewed data processing. In the process, the original splitting was dispensed with. Initially, a distinction was made between positive and negative PRA. The correlations together with RIP140 are now also shown in four-armed form.

  • Discussion : “In addition, similar results could be obtained in ovarian cancer.”  Citation needed.

The discussion was rewritten, this sentence was removed.

  • Methods: “We then correlated our results with the former staining of these samples for E6, LCor, RIP140, nuclear p53, H3K9ac and H3K4me3, which had recently been published” Correlation analysis with PR and RIP140 for some of those measures is not shown in the paper (H3K9ac, H3K4me3, P53,,,).

This passage was changed, what are now only parameters mentioned, which are listed in the results section.

11)  Conclusion: “The analyses of PR together with the transcription co-regulator RIP140 could act as a promising target not only for cervical cancer patients”   When saying “not only” please explain and give an example.

This sentence was rewritten, not only was removed.

12)  What is the possible biological explanations for PR levels being positive vs negative markers in different gynecological cancers?

In the recent workup, the nuclear expression pattern of RIP140 was shown to determine the prognostic significance of PRA, which is completely contrary even in the same tumor entity. It is possible that so far unknown influencing cofactors are the reason for the ambiguous effect of the progesterone receptor on different gynecological malignancies. 

13)  The immunohistochemistry was done with an antibody against PR-A but not against PR-B. It is, at least theoretically, possible that PR-B levels compensate for the PR-A levels which can affect the overall conclusions of this study. This should be addressed and mentioned as a possible limitation of the study.

      We are really thankful for this comment and added the sentence about PRB in the limitation of the study section:

“A limitation of our study might be that we only investigated the expression von PRA but not PRB in the samples. This seems to be an important starting point for further investigations.”

  1. Engelstaedter, V., B. Fluegel, S. Kunze, D. Mayr, K. Friese, U. Jeschke, and F. Bergauer. "Expression of the Carbohydrate Tumour Marker Sialyl Lewis a, Sialyl Lewis X, Lewis Y and Thomsen-Friedenreich Antigen in Normal Squamous Epithelium of the Uterine Cervix, Cervical Dysplasia and Cervical Cancer." Histol Histopathol 27, no. 4 (2012): 507-14.
  2. Kirn, V., I. Zaharieva, S. Heublein, F. Thangarajah, K. Friese, D. Mayr, and U. Jeschke. "Esr1 Promoter Methylation in Squamous Cell Cervical Cancer." Anticancer Res 34, no. 2 (2014): 723-7.
  3. Freier, C. P., A. Stiasny, C. Kuhn, D. Mayr, C. Alexiou, C. Janko, I. Wiest, U. Jeschke, and B. Kost. "Immunohistochemical Evaluation of the Role of P53 Mutation in Cervical Cancer: Ser-20 P53-Mutant Correlates with Better Prognosis." Anticancer Res 36, no. 6 (2016): 3131-7.
  4. Stiasny, A., C. Kuhn, D. Mayr, C. Alexiou, C. Janko, I. Wiest, U. Jeschke, and B. Kost. "Immunohistochemical Evaluation of E6/E7 Hpv Oncoproteins Staining in Cervical Cancer." Anticancer Res 36, no. 6 (2016): 3195-8.
  5. Beyer, S., J. Zhu, D. Mayr, C. Kuhn, S. Schulze, S. Hofmann, C. Dannecker, U. Jeschke, and B. P. Kost. "Histone H3 Acetyl K9 and Histone H3 Tri Methyl K4 as Prognostic Markers for Patients with Cervical Cancer." Int J Mol Sci 18, no. 3 (2017).
  6. Heidegger, H., S. Dietlmeier, Y. Ye, C. Kuhn, A. Vattai, C. Aberl, U. Jeschke, S. Mahner, and B. Kost. "The Prostaglandin Ep3 Receptor Is an Independent Negative Prognostic Factor for Cervical Cancer Patients." Int J Mol Sci 18, no. 7 (2017).
  7. Stiasny, A., C. P. Freier, C. Kuhn, S. Schulze, D. Mayr, C. Alexiou, C. Janko, I. Wiest, C. Dannecker, U. Jeschke, and B. P. Kost. "The Involvement of E6, P53, P16, Mdm2 and Gal-3 in the Clinical Outcome of Patients with Cervical Cancer." Oncol Lett 14, no. 4 (2017): 4467-76.
  8. Vattai, A., V. Cavailles, S. Sixou, S. Beyer, C. Kuhn, M. Peryanova, H. Heidegger, K. Hermelink, D. Mayr, S. Mahner, C. Dannecker, U. Jeschke, and B. Kost. "Investigation of Rip140 and Lcor as Independent Markers for Poor Prognosis in Cervical Cancer." Oncotarget 8, no. 62 (2017): 105356-71.
  9. Friese, K., B. Kost, A. Vattai, F. Marme, C. Kuhn, S. Mahner, C. Dannecker, U. Jeschke, and S. Heublein. "The G Protein-Coupled Estrogen Receptor (Gper/Gpr30) May Serve as a Prognostic Marker in Early-Stage Cervical Cancer." J Cancer Res Clin Oncol 144, no. 1 (2018): 13-19.
  10. Heublein, S., K. Friese, B. Kost, F. Marme, C. Kuhn, S. Mahner, C. Dannecker, D. Mayr, U. Jeschke, and A. Vattai. "Ta-Muc1 as Detected by the Fully Humanized, Therapeutic Antibody Gatipotzumab Predicts Poor Prognosis in Cervical Cancer." J Cancer Res Clin Oncol 144, no. 10 (2018): 1899-907.
  11. Kost, B. P., S. Beyer, L. Schroder, J. Zhou, D. Mayr, C. Kuhn, S. Schulze, S. Hofmann, S. Mahner, U. Jeschke, and H. Heidegger. "Glucocorticoid Receptor in Cervical Cancer: An Immunhistochemical Analysis." Arch Gynecol Obstet 299, no. 1 (2019): 203-09.
  12. Wang, Q., E. Schmoeckel, B. P. Kost, C. Kuhn, A. Vattai, T. Vilsmaier, S. Mahner, D. Mayr, U. Jeschke, and H. H. Heidegger. "Higher Ccl22+ Cell Infiltration Is Associated with Poor Prognosis in Cervical Cancer Patients." Cancers (Basel) 11, no. 12 (2019).
  13. Wang, Q., A. Steger, S. Mahner, U. Jeschke, and H. Heidegger. "The Formation and Therapeutic Update of Tumor-Associated Macrophages in Cervical Cancer." Int J Mol Sci 20, no. 13 (2019).
  14. Beilner, D., C. Kuhn, B. P. Kost, J. Juckstock, D. Mayr, E. Schmoeckel, C. Dannecker, S. Mahner, U. Jeschke, and H. H. Heidegger. "Lysine-Specific Histone Demethylase 1a (Lsd1) in Cervical Cancer." J Cancer Res Clin Oncol 146, no. 11 (2020): 2843-50.
  15. Dietlmeier, S., Y. Ye, C. Kuhn, A. Vattai, T. Vilsmaier, L. Schroder, B. P. Kost, J. Gallwas, U. Jeschke, S. Mahner, and H. H. Heidegger. "The Prostaglandin Receptor Ep2 Determines Prognosis in Ep3-Negative and Galectin-3-High Cervical Cancer Cases." Sci Rep 10, no. 1 (2020): 1154.
  16. Ye, Y., L. Peng, A. Vattai, E. Deuster, C. Kuhn, C. Dannecker, S. Mahner, U. Jeschke, V. von Schonfeldt, and H. H. Heidegger. "Prostaglandin E2 Receptor 3 (Ep3) Signaling Promotes Migration of Cervical Cancer Via Urokinase-Type Plasminogen Activator Receptor (Upar)." J Cancer Res Clin Oncol 146, no. 9 (2020): 2189-203.
  17. Beilner, D., C. Kuhn, B. P. Kost, T. Vilsmaier, A. Vattai, T. Kaltofen, S. Mahner, E. Schmoeckel, C. Dannecker, J. Juckstock, D. Mayr, U. Jeschke, and H. H. Heidegger. "Nuclear Receptor Corepressor (Ncor) Is a Positive Prognosticator for Cervical Cancer." Arch Gynecol Obstet 304, no. 5 (2021): 1307-14.
  18. Wang, Q., A. Vattai, T. Vilsmaier, T. Kaltofen, A. Steger, D. Mayr, S. Mahner, U. Jeschke, and H. Hildegard Heidegger. "Immunogenomic Identification for Predicting the Prognosis of Cervical Cancer Patients." Int J Mol Sci 22, no. 5 (2021).
  19. Beyer, S., M. Wehrmann, S. Meister, T. M. Kolben, F. Trillsch, A. Burges, B. Czogalla, E. Schmoeckel, S. Mahner, U. Jeschke, and T. Kolben. "Galectin-8 and -9 as Prognostic Factors for Cervical Cancer." Arch Gynecol Obstet 306, no. 4 (2022): 1211-20.
  20. Wang, Q., K. Sudan, E. Schmoeckel, B. P. Kost, C. Kuhn, A. Vattai, T. Vilsmaier, S. Mahner, U. Jeschke, and H. H. Heidegger. "Ccl22-Polarized Tams to M2a Macrophages in Cervical Cancer in Vitro Model." Cells 11, no. 13 (2022).

Round 2

Reviewer 1 Report

I really appreciate the improvements made by the authors as a result of my review. 

I suggest enriching the number of papers cited in the introductory part on the role of HPV in cervical cancer.

I indicate a very thorough literary review which, in my opinion, deserves to be mentioned (DOI: 10.3390/biology11081114)

  •  

Author Response

Reviewer 1:

I really appreciate the improvements made by the authors as a result of my review. 

I suggest enriching the number of papers cited in the introductory part on the role of HPV in cervical cancer.

I indicate a very thorough literary review which, in my opinion, deserves to be mentioned (DOI: 10.3390/biology11081114)

Thank you very much for your comment. The mentioned section has been underpinned with four more sources, including the proposed review.

Reviewer 3 Report

The authors answered all my questions and addressed my comments. 

Just two minor points:

1) It is not clear why on the Kaplan-Meier graph legends (figs 3-6) the authors say (example from fig 3) : "Statistical significance is shown as p-value from log-rank-test (p=0.012).  PRA=progesterone receptor A, IRS=immunoreactivity score. No statistics are computed because all cases are censored."
If there is statistical significance (p value etc) What does it mean "no statistics are computed..." ?
Is it related to some descriptive statistics ? please explain. Also "are censored" should be ""were censored".

2) lines -305-306: "we only investigated the expression von PRA but not PRB..."   I think von is a typo from German 

Author Response

Reviewer 3:

 The authors answered all my questions and addressed my comments. 

Just two minor points:

1) It is not clear why on the Kaplan-Meier graph legends (figs 3-6) the authors say (example from fig 3) : "Statistical significance is shown as p-value from log-rank-test (p=0.012).  PRA=progesterone receptor A, IRS=immunoreactivity score. No statistics are computed because all cases are censored."
If there is statistical significance (p value etc) What does it mean "no statistics are computed..." ?
Is it related to some descriptive statistics ? please explain. Also "are censored" should be ""were censored".

Thank you very much for your comment. For Figure 3 and 5, the missing statistics (survival times with SEM and CI) have been added accordingly. "No statistics are computed " was related to the means +-Confidence Interval of survival times for each group. The statistical evaluation has of course taken place - see p-value. For figure 4 and 6, this sentence has been changed to " Means +-Confidence Interval of survival times for each group are not computed because all cases were censored." accordingly.

2) lines -305-306: "we only investigated the expression von PRA but not PRB..."   I think von is a typo from German.

This was actually an excursion into German and has been improved accordingly.
